# SPARSE REFINEMENT FOR EFFICIENT HIGH-RESOLUTION SEMANTIC SEGMENTATION

## ABSTRACT

Semantic segmentation empowers numerous real-world applications, such as autonomous driving and augmented/mixed reality. These applications often operate on high-resolution images (*e.g.*, 8 megapixels) to capture the fine details. However, this comes at the cost of considerable computational complexity, hindering the deployment in latency-sensitive scenarios. In this paper, we introduce **SparseRefine**, a novel approach that enhances *dense low-resolution* predictions with *sparse high-resolution* refinements. Based on coarse low-resolution outputs, SparseRefine first uses an entropy selector to identify a sparse set of pixels with the least confidence. It then employs a *sparse* feature extractor to efficiently generate the refinements for those pixels of interest. Finally, it leverages a gated ensembler to apply these sparse refinements to the initial coarse predictions. SparseRefine can be seamlessly integrated into any existing semantic segmentation model, regardless of CNN- or ViT-based. SparseRefine achieves significant speedup: **1.5 to 3.9 times** when applied to HRNet-W48, SegFormer-B5, Mask2Former-T/L and SegNeXt-L on Cityscapes, with negligible to no loss of accuracy. We will release the code to reproduce our results. Our "*dense+sparse*" paradigm paves the way for efficient high-resolution visual computing.

## 1 INTRODUCTION

Semantic segmentation is a fundamental task within the field of computer vision and plays a crucial role in many real-world applications, such as autonomous driving and augmented/mixed reality. Over the past few years, advancements in deep neural networks (Long et al., 2015; Chen et al., 2017b; Wang et al., 2020; Xie et al., 2021; Cheng et al., 2022; Guo et al., 2022) have greatly enhanced the performance of semantic segmentation. However, the computational demands of these models present obstacles when it comes to their practical deployment on edge devices with limited resources.

Significant efforts have been dedicated to designing compact neural networks with reduced computational complexity (Howard et al., 2017; Sandler et al., 2018; Ma et al., 2018; Zhang et al., 2018; Iandola et al., 2016). However, in dense-prediction tasks like semantic segmentation, the image resolution makes greater impact to model's inference latency than the model size. This is because real-world segmentation applications often involve megapixel high-resolution images, which surpass the typical image classification workload by 1-2 orders of magnitude.

Reducing the image resolution through downsampling can result in a noticeable increase in speed. But, this comes at the cost of accuracy degradation. Segmentation models are generally more adversely affected by reduced resolution compared to classification models as low-resolution images result in the loss of fine details, including small or distant objects. The missing information can be safety-critical (*e.g.*, for autonomous driving).

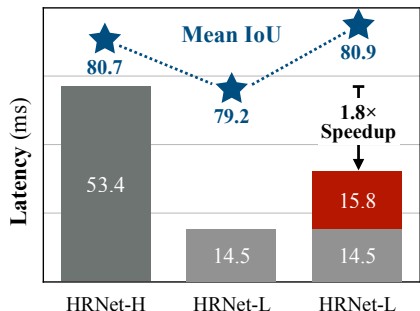

This paper introduces **SparseRefine** as a novel and complementary approach to address this problem. Instead of downsampling high-resolution inputs, SparseRefine enhances *dense low-resolution* predictions (based on downsampled inputs) with *sparse high-resolution* refinements. SparseRefine only refines a sparse set of carefully-selected pixels, enabling it to avoid

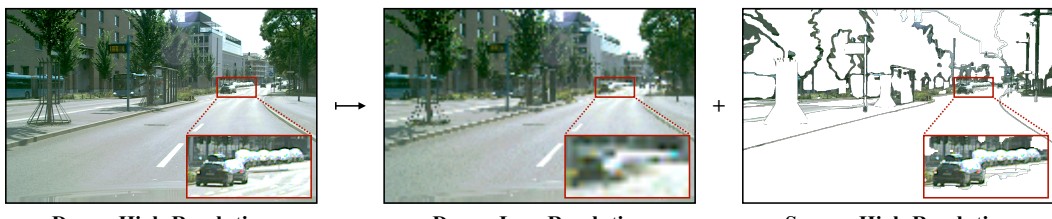

**Dense, High-Resolution**          **Dense, Low-Resolution**          **Sparse, High-Resolution**

Figure 1: Processing *dense high-resolution* inputs is computationally expensive. In this paper, we propose an alternative approach by integrating *dense low-resolution* and *sparse high-resolution* inputs, which provide complementary information about the overall scene layout and intricate object details. Leveraging the lower resolution and sparsity of these inputs allows for more efficient processing.

unnecessary high-resolution computations at the fine-grained pixel level. Also, it can be plugged into any existing semantic segmentation model, no matter it is CNN- or ViT-based. SparseRefine achieves remarkable and consistent speedup: **1.5 to 3.9 times** when applied to HRNet-W48, SegFormer-B5, Mask2Former-T/L and SegNeXt-L on Cityscapes, while maintaining accuracy. We will publicly release the code to facilitate further research and reproducibility.

## 2 RELATED WORK

**Semantic Segmentation** is a fundamental task in computer vision which assigns a class label to each pixel in an image. Following FCN (Long et al., 2015), early deep learning models (Badrinarayanan et al., 2017; Ronneberger et al., 2015) for semantic segmentation relied on CNN-based architectures. DeepLab and PSPNet (Zhao et al., 2017) improved FCN by introducing atrous convolution (Chen et al., 2015), spatial pyramid pooling (He et al., 2015; Zhao et al., 2017; Chen et al., 2016), encoder-decoder mechanism (Chen et al., 2017a), depthwise convolution (Chen et al., 2018) and neural architecture search (Liu et al., 2019). Follow-up research proposed attention mechanism (Fu et al., 2019) and object context modeling (Yuan et al., 2020). Recently, researchers also studied efficient segmentation architectures (Paszke et al., 2016; Wu et al., 2019; Poudel et al., 2019; Mehta et al., 2018; Yu et al., 2018; 2021; Zhao et al., 2018; Chen et al., 2020; Hong et al., 2021; Guo et al., 2022).

Recent advances in vision transformers (Dosovitskiy et al., 2021; Liu et al., 2021; 2022b; Wang et al., 2021b; Yuan et al., 2021a; Fan et al., 2021) also inspired the design of attention-based semantic segmentation models. SegFormer (Xie et al., 2021), SETR (Zheng et al., 2021), Segmenter (Strudel et al., 2021), HRFormer (Yuan et al., 2021b), SwinUNet (Cao et al., 2022) and EfficientViT (Cai et al., 2022) designed transformer-based backbones for segmentation, while MaskFormer (Cheng et al., 2021) and Mask2Former (Cheng et al., 2022) modeled semantic segmentation as mask classification.

**Activation Sparsity** naturally exists in videos (Pan et al., 2018; 2021a), point clouds (Graham et al., 2018; Liu et al., 2022a) and masked images in self-supervised visual pre-training (He et al., 2022; Gao et al., 2022; Tian et al., 2023; Huang et al., 2022). It can also be introduced through activation pruning (Pan et al., 2021b; Rao et al., 2021; Kong et al., 2022; Yin et al., 2021; Song et al., 2022; Liang et al., 2022), token merging (Bolya et al., 2023; Bolya & Hoffman, 2023) or clustering (Ma et al., 2023). These methods are specifically designed for classification or detection tasks, where there is no need to preserve information from all pixels. However, they are not suitable for semantic segmentation, which requires per-pixel predictions. An exception is SparseViT (Chen et al., 2023), which skips computation on pruned windows while retaining their features. As such, SparseViT also works for semantic segmentation tasks. We will demonstrate that SparseRefine achieves superior efficiency compared with SparseViT in Section 4. Recently, system and architecture researchers also created high-performance GPU libraries (Ren et al., 2018; Choy et al., 2019; Yan et al., 2018; Tang et al., 2022; 2023; Hong et al., 2023) and specialized hardware (Zhang et al., 2020; Wang et al., 2021a; Lin et al., 2021; Gondimalla et al., 2019; Wang et al., 2021c) to exploit activation sparsity.

**Mask Refinement** for segmentation has been studied even before the prevalence of deep learning. Traditional methods (Shi & Malik, 2000; Boykov et al., 2001; Felzenszwalb & Huttenlocher, 2004; Rother et al., 2004) formulated the task of semantic segmentation as graph cuts. The mask outputs were then post-processed using a conditional random field (CRF) (Lafferty et al., 2001; Blake et al., 2004; Krähenbühl & Koltun, 2011), which aimed to minimize energy and capture local consistency

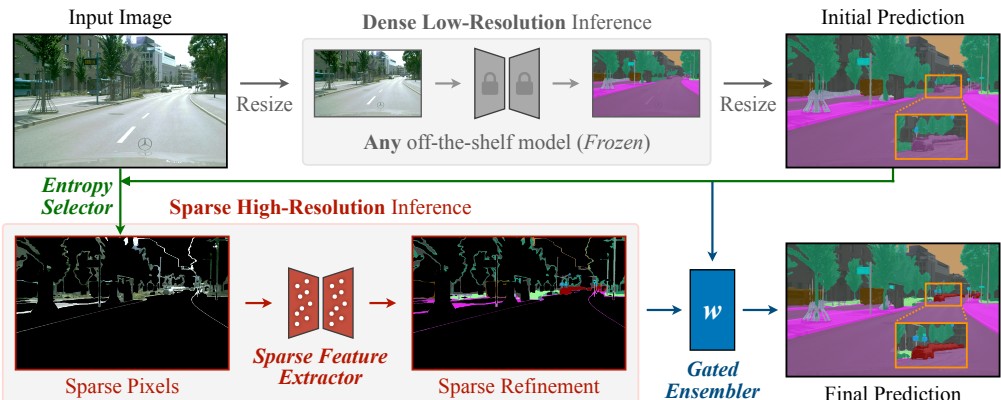

Figure 2: SparseRefine improves initial *dense low-resolution* predictions with *sparse high-resolution* refinements. It first uses an entropy selector to identify a sparse set of pixels with minimal confidence, and then employs a *sparse* feature extractor to efficiently generate refinements for those selected pixels. Afterwards, it applies these sparse refinements to the initial predictions with a gated ensembler.

in predicted labels. While CRF continues to impact the field in the deep learning era (Chen et al., 2016; Wu et al., 2018; Choy et al., 2019), its inefficiency eventually led to the development of PointRend (Kirillov et al., 2020) and RefineMask (Zhang et al., 2021). Inspired by graphics rendering, PointRend (Kirillov et al., 2020) first identifies uncertain pixels from deeper and lower resolution feature maps. These pixels are then refined using a PointNet (Qi et al., 2017a), leveraging interpolated shallower and higher resolution features. RefineMask (Zhang et al., 2021) gradually upsamples the predictions and incorporates the fine-grained features to alleviate the loss of details for high-quality instance mask prediction. Both PointRend and RefineMask upscale the *output resolution* with the help of high-resolution *features*, while SparseRefine is focused on reducing the *input resolution* and retains fine-grained details from full-scale *raw RGB pixels*. While PointRend and RefineMask prioritize improving *accuracy*, SparseRefine aims to minimize *latency*. Therefore, our method is fundamentally orthogonal to existing mask refinement strategies.

**Multi-Scale Models** have garnered popularity in high-resolution visual recognition tasks due to the diverse range of object sizes within an image. In early segmentation approaches, multi-resolution features were fused either using an FPN (Lin et al., 2017; Kirillov et al., 2019; Yu et al., 2018; 2021) or right before the prediction head (Chen et al., 2016; 2017a; He et al., 2015). Subsequently, new primitives such as OctaveConv (Chen et al., 2019), HRNet (Wang et al., 2020), and DDRNet (Wang et al., 2021c) were designed to more effectively leverage multi-scale features within the backbone. There have also been explorations on refining the predictions in a patch-wise manner (Verelst & Tuytelaars, 2022; Wu et al., 2020; Huang et al., 2019). Unlike SparseRefine, which enhances dense low-resolution predictions with *sparse* high-resolution details, existing methods focus on performing *dense* refinements. Also, while existing multi-scale models employ a *parallel* design for their low-resolution and high-resolution modules, SparseRefine adopts a *sequential* counterpart. This makes our method orthogonal to these designs. We will show in Section 4 that SparseRefine could bring further improvements to multi-scale models (*e.g.*, HRNet).

## 3 SPARSEREFINE

SparseRefine improves *dense low-resolution* predictions with *sparse high-resolution* refinements. As in Figure 2, it first uses an entropy selector to identify a sparse set of pixels with minimal confidence, and then employs a *sparse* feature extractor to efficiently generate refinements for those selected pixels. Afterwards, it applies these sparse refinements to the initial predictions with a gated ensembler. It is worth noting that while the individual modules we employ in our approach are well-established in the research community, integrating them together to support sparse refinement and achieve significant speedup in semantic segmentation is not a trivial task.

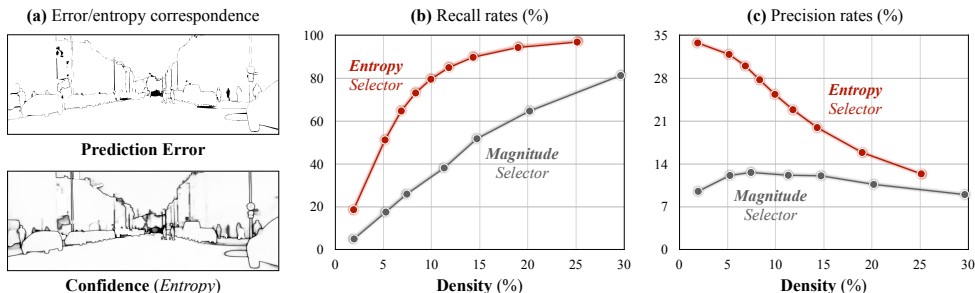

Figure 3: The entropy map exhibits a strong correlation with the error map **(a)**. Entropy selector has superior recall **(b)** and precision **(c)** rates than the magnitude selector across various density levels.

## 3.1 DENSE LOW-RESOLUTION PREDICTION

One of the most straightforward ways for accelerating inference is downsampling the input image. For instance, halving the resolution of HRNet-W48 (Wang et al., 2020) will lead to a $3.7\times$ speedup, close to the theoretical computation reduction of $4\times$. In this paper, we will use the coarse predictions from downsampled images as our starting point. All subsequent refinements will be built upon this.

As almost all semantic segmentation models inherently support different image resolutions, we do not need to make any modifications to the model architecture. The only adjustment that we need is to upsample the predictions to match the original input resolution (with nearest neighbour interpolation). Once the coarse predictions are obtained, there will be no further interaction with the original dense segmentation model (during both training and inference). As a result, our refinement module can be regarded as an add-on, allowing it to seamlessly support and enhance any off-the-shelf model.

## 3.2 SPARSE HIGH-RESOLUTION REFINEMENT

Low-resolution predictions are fast but not as accurate as high-resolution predictions. Fortunately, *the differences in their predictions primarily emerge in a sparse set of pixels*, often associated with small or distant objects and object boundaries. Building upon this observation, our objective is to *sparsely* refine the less accurate predictions so that we could bridge the accuracy gap while using only a portion of the reduced latency.

### 3.2.1 ENTROPY SELECTOR

The selection of sparse pixels plays a critical role in our entire pipeline as it directly determines the number and specific pixels on which we apply the refinement process. Ideally, we would want to choose those pixels that have been misclassified in the initial dense low-resolution prediction, but this is not feasible in practice. Inspired by recent works (Huang et al., 2019; Abdar et al., 2021) that utilize entropy maps to identify uncertain pixels, we adopt a similar thought and employ entropy as the criterion for selecting the pixels we need. Our intuition is that "*less confident predictions are more likely to be wrong*".

Visual verification from Figure 3a confirms a strong correspondence between the entropy map and the error map. Quantitatively, our entropy selector is able to identify around 80% of the misclassified pixels while selecting only 10% of the total pixels (Figure 3b). In contrast, the magnitude selector can only recover 40% of them with a similar density. The precision (Figure 3c) is less relevant in our case as the recall sets the accuracy upper bound for our refinement process. Apart from its effectiveness, our entropy selector is remarkably efficient, requiring only $\sim$2ms on an NVIDIA RTX 3090 GPU.

### 3.2.2 SPARSE FEATURE EXTRACTOR

After obtaining a set of sparse pixels from the entropy selector, our next step is then to generate the refinement for each of them. Processing sparse pixels poses great challenges due to their fundamentally different patterns compared to dense images. However, they do exhibit a similar pattern to 3D point clouds, where occlusion is frequently present, and only the geometry outline is captured. Despite being sparse, the pixels maintain a well-defined shape. The successful exploration

of point cloud segmentation in previous works (Wu et al., 2018; Choy et al., 2019; Tang et al., 2020) provides valuable insight indicating that sparse pixels should also contain contextual information that can support our sparse refinement approach.

In this paper, we utilize a modified version of MinkowskiUNet (Choy et al., 2019) as our *sparse feature extractor*. It follows the standard ResNet (He et al., 2016) basic block design with sparse convolutions and deconvolutions. Sparse convolution (Graham et al., 2018) is the sparse equivalent of conventional dense convolution with two main distinctions: firstly, sparse convolution avoids unnecessary computations for zero activations, and secondly, it preserves the same activation sparsity pattern throughout the model. These two properties make it much more efficient in processing our sparse pixel set. Furthermore, recent advances in system support for sparse convolution (Yan et al., 2018; Choy et al., 2019; Tang et al., 2022; Hong et al., 2023) enable us to translate the theoretical computational reduction, resulting from sparsity, into actual measured speedup. Please note that though we have chosen sparse convolution in this paper, alternative designs are also feasible, such as point-based convolutions (Qi et al., 2017b; Li et al., 2018; Wang et al., 2019) and more recent point cloud transformers (Fan et al., 2022; Sun et al., 2022; Liu et al., 2023; Wang et al., 2023).

The input to our sparse feature extractor is simply the raw RGB values of the selected pixels. We have explored adding more information from the low-resolution inference as input, such as final prediction or intermediate feature. While these additional features do contribute to faster convergence, they do not yield any improved performance. The output of our sparse feature extractor comprises multi-channel features for all selected pixels. We attach a simple linear classification head to generate the refinements for these pixels of interest.

### 3.2.3 GATED ENSEMBLER

After obtaining the refinement predictions, the straightforward approach is to directly substitute the initial predictions at the corresponding pixels. However, this approach is not always optimal. This is because, compared to dense pixels, the context information available for sparse pixels in high-resolution is relatively limited. Incorporating initial predictions from dense low-resolution images, which provide more comprehensive context information, can be beneficial.

We introduce the *gated ensembler* to intelligently integrate the initial predictions ($y_1$) and the refined predictions ($y_2$). The key idea is to generate a weighting factor $w \in [0, 1]$ for each pixel of interest and utilize it to fuse the two predictions. Concretely, the final predictions are generated by

$$y = f(w \cdot y_1 + (1 - w) \cdot y_2), \quad \text{where } w = \text{sigmoid}(g([y_1; y_2; e_1; e_2])). \tag{1}$$

Here, $f(\cdot)$ and $g(\cdot)$ are two-layer multi-layer perceptrons (MLPs). To generate the weighting factor, we provide both the raw predictions ($y_{1,2}$) and their corresponding entropies ($e_{1,2}$) as inputs to $g$.

## 4 EXPERIMENTS

### 4.1 SETUP

**Dataset.** We conduct our primary experiments on Cityscapes (Cordts et al., 2016), which comprises 5,000 high-resolution images of urban scenes. Each image has a resolution of $1024 \times 2048$ and is labeled with 19 categories for semantic segmentation. We also validate the general effectiveness of our method on three additional datasets: BDD100K for autonomous driving (Yu et al., 2020), DeepGlobe for aerial images (Demir et al., 2018), and ISIC for medical images (Codella et al., 2018). Our main evaluation criterion is the mean intersection over union (mIoU).

**Baselines.** To demonstrate that our method is capable of generalizing to different model architectures, we use five different models encompassing both convolution-based and transformer-based architectures. We choose HRNet-W48 (Wang et al., 2020) as the convolution-based baseline, and SegFormer-B5 (Xie et al., 2021), Mask2Former-T (Cheng et al., 2022), Mask2Former-L (Cheng et al., 2022), and SegNeXt-L (Guo et al., 2022) as our transformer-based baselines. We reproduce the results of all high-resolution and low-resolution baselines using MMSegmentation v1.0.0 (Contributors, 2020). We follow the default training settings, with minimal modifications made to the data augmentation parameters to accommodate the lower resolution. Also, as the results for Mask2Former exhibit some instability, we report the mean of three runs for a more reliable assessment.

| | Input Resolution | #Params (M) | #MACs (T) | Latency (ms) | Mean IoU |
|---|---|---|---|---|---|
| HRNet-W48 | 1024×2048 (D) | 65.9 | 0.75 | 53.4 | 80.7 |
| HRNet-W48 | 512×1024 (D) | 65.9 | 0.19 (2.3×) | 14.5 (1.8×) | 79.2 (+0.2) |
| + **SparseRefine** (Ours) | 1024×2048 (S) | 85.7 | 0.32 | 30.3 | 80.9 |
| SegFormer-B5 | 1024×2048 (D) | 82.0 | 1.16 | 140.6 | 81.1 |
| SegFormer-B5 | 512×1024 (D) | 82.0 | 0.17 (3.6×) | 18.5 (3.9×) | 78.7 (+0.1) |
| + **SparseRefine** (Ours) | 1024×2048 (S) | 101.8 | 0.32 | 36.2 | 81.2 |
| Mask2Former-T | 1024×2048 (D) | 36.7 | 0.62 | 66.8 | 81.1 |
| Mask2Former-T | 512×1024 (D) | 36.7 | 0.16 (1.6×) | 19.1 (1.5×) | 78.6 (+0.2) |
| + **SparseRefine** (Ours) | 1024×2048 (S) | 56.5 | 0.39 | 44.8 | 81.3 |
| Mask2Former-L | 1024×2048 (D) | 207.0 | 1.99 | 150.8 | 83.0 |
| Mask2Former-L | 512×1024 (D) | 207.0 | 0.51 (2.4×) | 45.4 (1.8×) | 80.9 (+0.0) |
| + **SparseRefine** (Ours) | 1024×2048 (S) | 226.8 | 0.84 | 84.4 | 83.0 |
| SegNeXt-L | 1024×2048 (D) | 48.8 | 0.53 | 86.3 | 83.0 |
| SegNeXt-L | 640×1280 (D) | 48.8 | 0.21 (1.7×) | 33.6 (1.8×) | 80.8 (−0.2) |
| + **SparseRefine** (Ours) | 1024×2048 (S) | 68.6 | 0.32 | 49.1 | 82.8 |

Table 1: SparseRefine effectively closes the accuracy gap between low-resolution and high-resolution predictions, achieving a remarkable reduction in computational cost by **1.6 to 3.6 times** and inference latency by **1.5 to 3.9 times**. In this table, (D) and (S) denote dense and sparse inputs, respectively.

| | | BDD100K | | DeepGlobe | | ISIC | |
|---|---|---|---|---|---|---|---|
| | Resolution | Latency (ms) | mIoU | Latency (ms) | mIoU | Latency (ms) | mIoU |
| HRNet-W48 | Full (D) | 23.5 | 63.6 | 146.4 | 73.4 | 157.7 | 82.3 |
| HRNet-W48 | Half (D) | 6.1 (1.5×) | 60.7 (−0.1) | 38.7 (1.6×) | 72.9 (+0.0) | 40.6 (2.0×) | 80.8 (+0.2) |
| + **SparseRefine** | Full (S) | 15.6 | 63.5 | 92.9 | 73.4 | 79.4 | 82.5 |

Table 2: SparseRefine effectively generalizes across autonomous driving (BDD100K), aerial (Deep-Globe) and medical (ISIC) datasets, achieving a **1.5-2.0×** measured speedup with no loss of accuracy.

**Model Details.**  We set a different entropy threshold for each model in our entropy selector. Our sparse feature extractor is a modified MinkowskiUNet that has five stages with channel dimensions of 32, 64, 128, 256, and 512 for each stage. At each stage, there are two ResNet basic blocks before downsampling and another two after upsampling. Our gated ensembler employs two linear layers to produce the weighting factor and an additional two layers to combine the predictions, both with a hidden dimension of 64. We refer the readers to the appendix for more implementation details.

**Training Details.**  SparseRefine is trained independently from the dense baselines. We use the same data augmentation and training strategy employed by the dense baseline to ensure that performance improvements stem solely from our method. For data augmentation, we apply standard techniques such as random scaling (between 0.5 and 2.0), horizontal flipping, cropping (with a size of 512×1024), and photometric distortion. We apply the standard cross entropy loss to supervise the model. We adopt AdamW (Loshchilov & Hutter, 2019) as our optimizer, with an initial learning rate of 0.0003 and a weight decay of 0.05. We gradually decay the learning rate following the cosine-annealing schedule (Loshchilov & Hutter, 2017). We train the model for 500 epochs with a batch size of 32. The training takes around 12 hours on 8 NVIDIA RTX A6000 GPUs.

**Latency Details.**  We use cuBLAS (NVIDIA) for all dense operations and TorchSparse v2.0 (Tang et al., 2022; 2023) for all sparse operations. We measure the inference latency of all methods using a single NVIDIA RTX 3090 GPU with FP16 precision and a batch size of 4. We provide additional results for different precisions and batch sizes in the appendix. We omit all batch normalization layers for latency measurement as they can be folded into their preceding convolution layers. We report the average latency over 500 inference steps, with an additional 100 steps designated as warm-up.

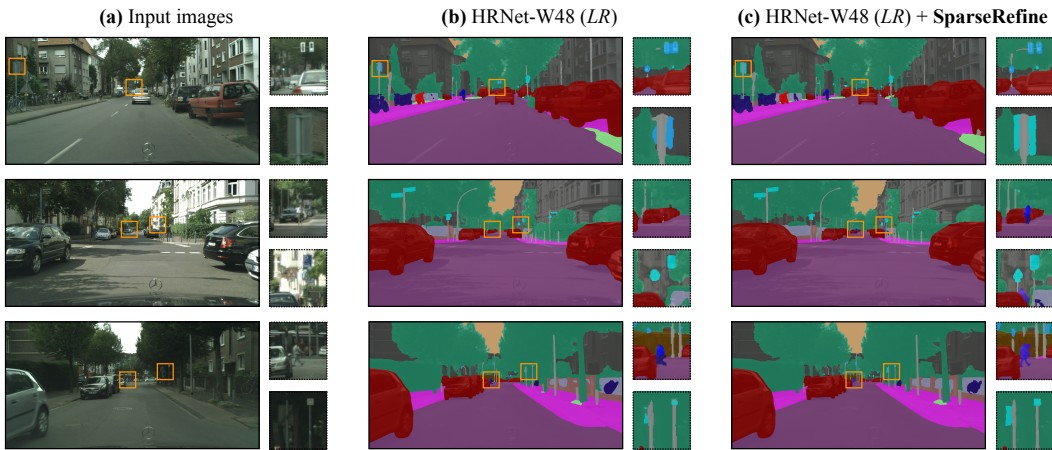

Figure 4: SparseRefine improves the low-resolution (*LR*) baseline with substantially better recognition of small, distant objects and finer detail around object boundaries.

|  | Latency (ms) | Mean IoU |  | Latency (ms) | Mean IoU |
|---|---|---|---|---|---|
| Mask2Former-L | 150.8 | 83.0 | HRNet-W48 | 53.4 | 80.7 |
| + SparseViT | 132.3 | 83.2 | + PointRend | 30.5 | 79.8 |
| + **SparseRefine** (Ours) | 84.4 ⌡1.6× | 83.0 | + **SparseRefine** (Ours) | 30.3 | 80.9 ⌡+1.1 |

| (a) Comparison to token pruning | (b) Comparison to mask refinement |
|---|---|

Table 3: SparseRefine is more efficient/effective than token pruning and mask refinement approaches.

## 4.2 RESULTS

We present our key experimental results in Table 1. It is evident that SparseRefine achieves substantial improvements in terms of #MACs and latency, while maintaining competitive or even better accuracy compared to the baselines. Specifically, SparseRefine accelerates the baselines by at least **1.5 times** and reduces the MACs by at least **1.6 times**. Notably, SparseRefine achieves a significant speedup of **3.9 times** for SegFormer-B5. This huge improvement can be attributed to the fact that SegFormer incorporates a vanilla self-attention module with high computational complexity ($O(H^2W^2)$), and our "*downsample then sparsely refine*" strategy in SparseRefine can drastically reduce the computational cost via downsampling while recovering the accuracy through refining. From Table 2, SparseRefine demonstrates effective generalization across driving (BDD100K), aerial (DeepGlobe), and medical (ISIC) datasets. It delivers a consistent speedup from **1.5** to **2.0** times without compromising accuracy.

In addition to the quantitative results, we also present qualitative results in Figure 4. In the middle column, it can be observed that the low-resolution baseline struggles to accurately classify pixels in distant areas and often misclassifies details near the edges. Our SparseRefine significantly improves the ambiguous predictions, as shown in the third column. The second row is a notable example, where the segmentation on low-resolution images fails to detect a person in the far distance, while SparseRefine accurately predicts their presence. Furthermore, SparseRefine even achieves accurate predictions in challenging cases, such as the thin rod of traffic lights in the first row. We provide more visualizations in the appendix. These results further demonstrate the effectiveness of our method.

**Comparison to Token Pruning.** SparseViT (Chen et al., 2023) is one of the first works that enables the feasibility of applying token pruning on dense prediction tasks, such as semantic segmentation. As in Table 3a, SparseRefine exhibits a notable advantage over SparseViT in terms of latency reduction, achieving a 1.6× speedup while maintaining comparable accuracy. SparseRefine is fundamentally orthogonal to these token pruning methods and can potentially be applied in conjunction with them.

**Comparison to Mask Refinement.** We also compare SparseRefine with PointRend (Kirillov et al., 2020), a mask refinement method. To ensure fair comparisons, we adjust the images to a resolution of 672×1344 for PointRend, aligning its latency with our SparseRefine. As in Table 3b, PointRend experiences a notable performance decline, while our method demonstrates an improvement over

| Criteria | Density | Precision | Recall | mIoU |
|---|---|---|---|---|
| Random | 10.0% | 0.3% | 10.0% | 79.2 |
| Magnitude | 11.3% | 12.1% | 38.1% | 80.2 |
| Entropy | 11.8% | **22.8%** | **84.9%** | **80.9** |
| Oracle | 3.3% | 100% | 100% | 92.8 |

| $\alpha$ | Density | Recall | Latency | mIoU |
|---|---|---|---|---|
| 0.8 | 3.4% | 32.2% | 23.1 ms | 79.9 |
| 0.6 | 6.9% | 64.6% | 27.0 ms | 80.4 |
| 0.3 | 11.8% | 84.9% | 30.3 ms | 80.9 |
| 0.1 | 19.0% | 94.3% | 37.3 ms | 81.1 |

| Strategy | mIoU |
|---|---|
| Direct | 77.7 |
| Entropy | 80.3 |
| Gated | **80.9** |
| Oracle | 85.3 |

(a) Entropy is a more effective indicator for identifying misclassified pixels than feature magnitude.

(b) Performance improves with more pixels kept, but latency also increases.

(c) Gated ensembler is the best.

Table 5: Ablation experiments to validate our design choices. Default settings are marked in blue.

| | #MACs | Latency |
|---|---|---|
| Entropy Selector | 0 | 2.3 ms |
| Sparse Feature Extractor | 0.129T | 10.9 ms |
| Gated Ensembler | 0.001T | 2.2 ms |

| Backend | Activation | Latency |
|---|---|---|
| cuBLAS | Dense | 52.0 ms |
| SpConv v2.3.5 | Sparse | 14.2 ms |
| TorchSparse v2.1.0 | Sparse | **10.9 ms** |

Table 6: **Breakdown of #MACs and latency**. Entropy selector and gated ensembler are lightweight.

Table 7: **Sparse inference backend**. Sparse inference is more efficient than dense inference.

the baseline. PointRend relies on an MLP-based mask refinement approach using hidden features. However, when low-resolution images are used as input, the refinement process struggles to effectively compensate for information loss caused by downsampling. In contrast, SparseRefine operates directly on the high-resolution image, mitigating information loss and ultimately enhancing performance.

**Comparison to Patch Refinement.** Some existing methods (Huang et al., 2019; Wu et al., 2020; Verelst & Tuytelaars, 2022) refine the prediction in a *coarse-grained patch* level, while SparseRefine refines the prediction in a *fine-grained pixel* level. As depicted in Figure 3a, errors tend to be *scattered sparsely* across the entire image, making fine-grained sparsity a more suitable solution. Patch-based refinement can often lead to substantial redundant computation, as not every pixel within a patch may need refinement. This inefficiency renders the patch-based methods less effective. From Table 4, SparseRefine outperforms patch refinement baselines (with a patch size of 256 or 512), achieving a speedup of **1.5** to **1.8** times while also delivering higher accuracy.

| | Latency (ms) | mIoU |
|---|---|---|
| Baseline | 53.4 | 80.7 |
| PatchRefine (512) | 44.1 | 80.8 |
| PatchRefine (256) | 55.4 | 80.8 |
| SparseRefine | **30.3** | **80.9** |

Table 4: Sparse refinement is faster and more accurate than patch refinement.

## 4.3 ANALYSIS

In this section, we analyze various alternative designs for the components of our method. Additionally, we provide detailed breakdowns of the improvements in both accuracy and efficiency. All the analyses in this section are conducted based on HRNet-W48 as the baseline model.

### 4.3.1 ALTERNATIVE DESIGNS

**Pixel Selector.** We compare our proposed entropy-based pixel selector to other alternatives, including random selector and magnitude selector, as shown in Table 5a. The random selector randomly selects pixels based on a density hyperparameter that we set. Compared to the entropy-based selector, the random selector shows a substantial performance drop of 1.7 mIoU, failing to achieve any improvement over the low-resolution baseline. This decline can be attributed to the notably low recall rate (10%) of the random selection approach and its lack of principled criteria to ensure the selection of misclassified pixels.

Another alternative is the magnitude-based selector, which employs the L2 magnitude activation as the importance score for pixel selection. As depicted in Table 5a, it is evident that the magnitude-based selector still exhibits significantly lower recall and precision compared to the entropy-based selector. Consequently, the magnitude-based selector performs worse than our entropy selector by 0.7 mIoU.

Furthermore, we also showcase the performance of the oracle setting, wherein we select incorrect predictions solely based on the ground truth. This highlights the considerable room for improvement and underscores the immense potential of our proposed paradigm.

| | Road | Sidewalk | Building | Wall | Fence | Pole | Traffic Light | Traffic Sign | Vegetation | Terrain | Sky | Person | Rider | Car | Truck | Bus | Train | Motorcycle | Bicycle | mIoU |
|---|---|---|---|---|---|---|---|---|---|---|---|---|---|---|---|---|---|---|---|---|
| HRNet-W48 (512×1024) | 98.3 | 86.1 | 92.8 | **57.8** | **66.8** | 65.7 | 70.0 | 78.9 | 92.4 | **64.7** | 94.8 | 81.1 | 62.3 | 95.0 | 84.3 | 88.9 | 82.6 | 64.8 | 77.4 | 79.2 |
| **+ SparseRefine** (Ours) | **98.4** | **86.7** | **93.4** | 57.7 | 66.7 | **70.5** | **75.0** | **82.0** | **93.0** | 64.4 | **95.4** | **84.3** | **67.4** | **95.8** | **85.6** | **89.6** | **83.3** | **67.9** | **80.1** | **80.9** |
| HRNet-W48 (1024×2048) | 98.4 | 86.6 | 93.2 | 55.7 | 64.9 | 71.5 | 75.8 | 82.9 | 92.8 | 65.4 | 95.4 | 84.6 | 65.8 | 95.7 | 80.4 | 91.5 | 83.2 | 70.1 | 80.1 | 80.7 |

Table 8: SparseRefine consistently improves the performance of the low-resolution baseline across different categories, particularly for small objects.

**Entropy Threshold.** We present an analysis of the impact of different entropy thresholds on latency and accuracy, as outlined in Table 5b. In essence, the entropy threshold involves a trade-off between latency and accuracy: a lower entropy threshold leads to the selection and refinement of more pixels, resulting in improved performance but increased latency. We select a moderate setting with $\alpha = 0.3$ for HRNet-W48, which matches the accuracy of the high-resolution baseline with the largest speedup. The optimal $\alpha$ for different models could be different. We provide more details in the appendix.

**Ensembler.** We investigate different ensemble strategies in Table 5e. The simplest approach is to directly replace the initial predictions with the refined predictions. However, this is suboptimal due to SparseRefine's limited context (discussed in Section 3.2.3). Another alternative is the entropy-based ensembler that compares the entropy before and after refinement to determine which predictions to choose. In comparison, our gated ensembler offers a softer and more compact way to incorporate refinement into the prediction. It is noteworthy that the gated ensembler outperforms the entropy-based ensembler by 0.6 mIoU. Additionally, we analyze the performance of the oracle setting, where we choose the better of the predictions before and after refinement. This analysis reveals a substantial room for improvement of 4.4 mIoU, further emphasizing the potential for future enhancements.

### 4.3.2 BREAKDOWNS

We begin by analyzing the per-class performance, as presented in Table 8. It is evident that SparseRefine consistently improves the performance of the low-resolution baseline in almost every category, particularly for small instances such as person, rider, pole, *etc*. These categories also exhibit the most significant degradation in the low-resolution baseline when compared to the high-resolution baseline. This observation highlights the effectiveness of SparseRefine in capturing fine-grained details, thanks to its ability to utilize sparse high-resolution information.

The breakdown of #MACs and latency for each component is presented in Table 6. It can be observed that the entropy selector and gated ensembler contribute minimally to the overall computation, while the feature extractor accounts for the majority. We have implemented the sparse feature extractor using different inference backends. As shown in Table 7, our input activation has high (approximately 90%) sparsity. Therefore, sparse inference backends like SpConv and TorchSparse are more suitable than dense inference backends such as cuBLAS.

## 5 CONCLUSION

We present SparseRefine that enhances low-resolution dense predictions with high-resolution sparse refinements. It first incorporates an entropy selector to identify a sparse set of pixels with the lowest confidence, followed by a sparse feature extractor that efficiently generates refinements for those selected pixels. Finally, a gated ensembler is utilized to integrate these sparse refinements with the initial coarse predictions. Notably, SparseRefine can be seamlessly integrated into various existing semantic segmentation models, irrespective of their model architectures. Empirical evaluation on the Cityscapes dataset demonstrated remarkable speed improvements, achieving measured speedups of **1.5 to 3.9 times** when applied to HRNet-W48, SegFormer-B5, Mask2Former-T/L and SegNeXt-L, with negligible to no loss of accuracy. We believe that the speedups that accrue from our approach of combining *low-resolution* prediction followed by *sparse high-resolution* refinement, will further enable the deployment of high-resolution semantic segmentation in latency-sensitive applications.

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
