## A1 MORE DETAILS

The detailed architecture of our sparse feature extractor is illustrated in Figure A1. This extractor is structured into a total of 5 stages. Channel dimensions for each stage are defined as 32, 64, 128, 256, and 512, respectively. At each stage, we employ two ResNet basic blocks before downsampling and an additional two after upsampling. These ResNet basic blocks incorporate sparse convolution, with a kernel size of 3 used for each sparse convolution operation. This choice enables us to capture ample spatial information while maintaining a manageable model size.

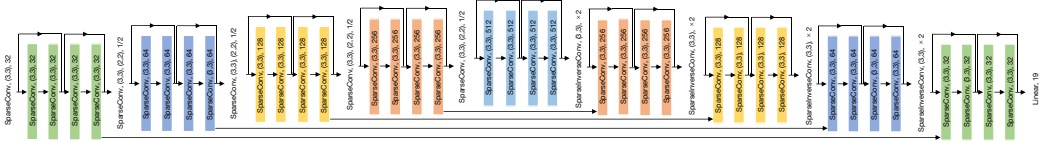

Figure A1: Detailed model architecture of sparse feature extractor.

Different models often exhibit distinct entropy distributions, a phenomenon that may be attributed to differences in model architecture, loss formulation, and training settings. These differences necessitate the selection of unique thresholds for each model. In this work, we determine a threshold for each model to target a recall rate at least 80%. This step requires the calculation of the density-recall curve, illustrated in Figure 3b of the main paper. The process is both efficient and convenient, as it avoids the need for repeatedly retraining the model through a trial-and-error method. We can effectively determine the entropy threshold without retraining the model many times. The entropy thresholds for different models are presented in Table A1.

|  | Entropy | Recall | Precision | Density |
|---|---|---|---|---|
| HRNet-W48 Wang et al. (2020) | 0.3 | 84.9% | 22.8% | 11.8% |
| SegFormer-B5 (Xie et al., 2021) | 0.1 | 84.7% | 20.1% | 13.9% |
| Mask2Former-T (Cheng et al., 2022) | 0.05 | 93.4% | 13.1% | 24.1% |
| Mask2Former-L (Cheng et al., 2022) | 0.005 | 97.3% | 9.1% | 35.0% |
| SegNeXt-L (Guo et al., 2022) | 0.1 | 80.8% | 24.2% | 10.0% |

Table A1: Entropy threshold, recall rate, precision, and density for each baseline model.

## A2 MORE RESULTS

In the main paper, we reported the latency of SparseRefine using FP16 precision and a batch size of 4 on an NVIDIA RTX 3090. To provide a more comprehensive understanding of latency, Table A2 offers a detailed measurement of latency across various precision modes and batch sizes. It's noteworthy that SparseRefine consistently reduces latency across different precision modes and batch sizes. As the batch size increases, we generally observe a decrease in the inference latency of SparseRefine. This phenomenon can be attributed to larger batch sizes leading to higher GPU utilization during sparse convolution operations. Furthermore, it's worth mentioning that some larger models encounter out-of-memory errors when performing inference on high-resolution inputs with a large batch size. In contrast, SparseRefine does not face this issue, underscoring its efficiency and robustness in handling resource constraints.

We also conducted latency evaluations of SparseRefine on an NVIDIA Jetson AGX Orin, a widely adopted platform for autonomous driving, and the outcomes are summarized in Table A3. These results offer crucial insights into the practical applicability of our approach. In general, it is observed that the inference latency on the NVIDIA Jetson AGX Orin platform is roughly four times slower compared to the NVIDIA RTX 3090. Despite this discrepancy, SparseRefine manages to achieve a noteworthy speedup on the NVIDIA Jetson AGX Orin, affirming its promise and suitability for real-world applications in the context of autonomous driving.

| | | FP16 | | | | FP32 | | | |
|---|---|---|---|---|---|---|---|---|---|
| | Input Resolution | B1 | B2 | B4 | B8 | B1 | B2 | B4 | B8 |
| HRNet-W48 | 1024×2048 (D) | 56.3 | 54.1 | 53.4 | 51.5 | 95.2 | 96.5 | 90.9 | 89.7 |
| HRNet-W48 | 512×1024 (D) | 20.2 | 15.1 | 14.5 | 14.1 | 26.8 | 25.8 | 24.4 | 23.8 |
| + SparseRefine | 1024×2048 (S) | 40.2 | 31.8 | 30.3 | 28.6 | 65.4 | 60.5 | 57.3 | 55.3 |
| SegFormer-B5 | 1024×2048 (D) | 149.5 | 141.1 | 140.6 | 141.3 | 311.5 | 302.1 | 299.2 | OOM |
| SegFormer-B5 | 512×1024 (D) | 39.0 | 20.6 | 18.5 | 17.8 | 43.7 | 39.9 | 38.3 | 37.2 |
| + SparseRefine | 1024×2048 (S) | 61.0 | 39.5 | 36.2 | 34.0 | 86.6 | 78.5 | 74.9 | 72.5 |
| Mask2Former-T | 1024×2048 (D) | 73.9 | 72.7 | 66.8 | 65.4 | 156.6 | 145.9 | 138.8 | OOM |
| Mask2Former-T | 512×1024 (D) | 37.5 | 24.1 | 19.1 | 17.1 | 49.7 | 42.1 | 37.6 | 34.6 |
| + SparseRefine | 1024×2048 (S) | 67.9 | 51.7 | 44.8 | 44.4 | 111.9 | 99.2 | 93.7 | 89.3 |
| Mask2Former-L | 1024×2048 (D) | 158.4 | 150.8 | 146.3 | 144.6 | 382.2 | 371.1 | 367.1 | OOM |
| Mask2Former-L | 512×1024 (D) | 56.3 | 45.4 | 40.3 | 37.5 | 113.0 | 100.8 | 94.6 | 92.1 |
| + SparseRefine | 1024×2048 (S) | 99.8 | 84.4 | 74.9 | 71.3 | 198.5 | 181.5 | 173.5 | 169.6 |
| SegNeXt-L | 1024×2048 (D) | 85.3 | 87.3 | 86.3 | 84.0 | 132.7 | 155.2 | 141.5 | 134.2 |
| SegNeXt-L | 640×1280 (D) | 34.7 | 35.4 | 33.6 | 32.8 | 53.5 | 62.7 | 56.7 | 52.7 |
| + SparseRefine | 1024×2048 (S) | 55.1 | 51.8 | 49.1 | 47.0 | 88.5 | 94.2 | 86.0 | 80.9 |

Table A2: Latency under different precision and batch size on NVIDIA RTX 3090. SparseRefine benefits from high utilization of sparse convolution, particularly as the batch size increases.

| | | FP16 | | | FP32 | | |
|---|---|---|---|---|---|---|---|
| | Input Resolution | B1 | B2 | B4 | B1 | B2 | B4 |
| HRNet-W48 | 1024×2048 (D) | 233.0 | 217.1 | 211.1 | 360.6 | 365.8 | 349.4 |
| HRNet-W48 | 512×1024 (D) | 78.6 | 63.9 | 60.5 | 98.9 | 93.6 | 90.5 |
| + SparseRefine | 1024×2048 (S) | 157.0 | 138.9 | 134.1 | 277.4 | 269.9 | 265.3 |
| SegFormer-B5 | 1024×2048 (D) | 751.5 | 721.0 | 719.5 | 1220.5 | 1172.5 | 1172.8 |
| SegFormer-B5 | 512×1024 (D) | 110.8 | 102.4 | 98.8 | 179.9 | 161.3 | 163.1 |
| + SparseRefine | 1024×2048 (S) | 197.5 | 186.7 | 183.1 | 386.1 | 363.7 | 362.3 |
| Mask2Former-T | 1024×2048 (D) | 449.1 | 433.7 | 422.8 | 583.2 | 591.7 | 585.6 |
| Mask2Former-T | 512×1024 (D) | 133.0 | 126.2 | 115.1 | 174.8 | 171.4 | 149.8 |
| + SparseRefine | 1024×2048 (S) | 259.0 | 250.5 | 241.0 | 486.1 | 479.1 | 453.7 |
| Mask2Former-L | 1024×2048 (D) | 882.1 | 903.2 | 888.1 | 1217.7 | 1198.5 | 1201.8 |
| Mask2Former-L | 512×1024 (D) | 262.2 | 243.7 | 226.6 | 356.7 | 334.0 | 317.6 |
| + SparseRefine | 1024×2048 (S) | 432.6 | 412.0 | 396.9 | 784.4 | 757.5 | 736.9 |
| SegNeXt-L | 1024×2048 (D) | 525.4 | 529.3 | 539.1 | 712.0 | 743.7 | 731.8 |
| SegNeXt-L | 640×1280 (D) | 209.9 | 214.6 | 202.7 | 280.3 | 296.9 | 288.5 |
| + SparseRefine | 1024×2048 (S) | 278.4 | 279.5 | 266.2 | 447.5 | 450.6 | 441.5 |

Table A3: Latency under different precision and batch size on NVIDIA Jetson AGX Orin.

## A3 MORE VISUALIZATIONS

We have included additional visualization results in Figure A2. These supplementary results exhibit enhanced recognition of small, distant objects and finer details around object boundaries. These observations serve as further evidence of the efficacy of our SparseRefine.

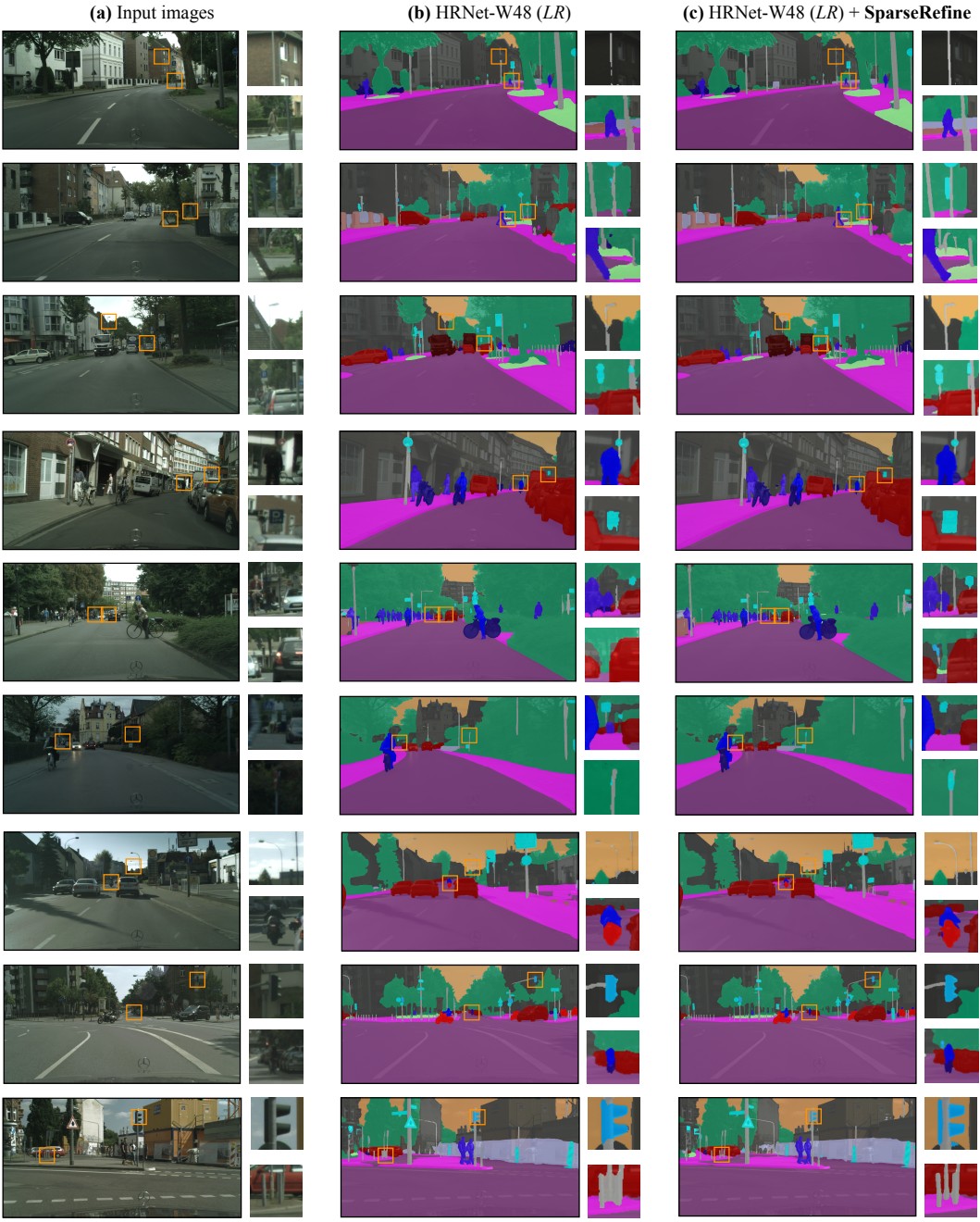

Figure A2: SparseRefine improves the low-resolution (*LR*) baseline with substantially better recognition of small, distant objects and finer detail around object boundaries.