# OpenReview forum: "Sparse Refinement for Efficient High-Resolution Semantic Segmentation"
_ICLR.cc/2024/Conference — Submitted to ICLR 2024_

### Official Review · Reviewer_Jr1p · 2023-10-14

**Soundness:** 3 good
**Presentation:** 4 excellent
**Contribution:** 3 good
**Rating:** 5
**Confidence:** 5

**Summary:**

To improve semantic segmentation performance on low resolution images, the authors propose Sparse Refinement, which composes of entropy selector, 3d-point-clouds-encoder-like sparse feature extractor, and ensembler. As shown in the experiments, the proposed plug-to-play module speed up sota models while keeping the prediction accuracy. The proposed method is also compared to token pruning and mask refinement. Extensive designs are discussed and breakdowns are shown with clarity.

**Strengths:**

The paper is well written and easy to follow. The motivation and the proposed module is described clearly. Experiments and ablation studies are conducted to verify its superiority. Although the proposed module seems to be a simple collection of existing methods, the performance improvement it brings is significant. The introduction of sparse feature extractor, which is like MinkowskiUNet, extracts features from sparse patterns. The entropy selector is also verifies in figure 3.

**Weaknesses:**

The proposed module is powerful and clearly stated. The main weakness in this paper I find is the lack of comprehensive experiments.
1. To evaluate a deep learning based model, the authors may want to try various datasets to prove its generalization. The cityscapes, and three domain-specific datasets are not enough. Please try coco, ade20k,pascal…
2. For table 2, it would be better if the authors briefly describe evaluation metrics.
3. In ablation studies, how does the model without ensembler performs?
4. Similar to weakness1, the authors may want to discuss how does the entropy selector behaves on other datasets.
5. The authors may want to briefly discuss, or derive mathematically, why final prediction or intermediate feature doesn’t improve performance.

**Questions:**

1. in figure 3, is the recall-precision curve based on the sample image or the cityscapes dataset as a whole?
2. The authors don’t have to answer this question if their time is limited: since the sparse feature extractor is inspired by 3d point clouds’ encoders, I wonder how does the proposed module perform on 3d benchmarks like 3d semantic segmentation and depth estimation.

**Details Of Ethics Concerns:**

I don't have ethnic concerns on this paper.

---

> ### Author Response · Authors · 2023-11-18
>
> Thank you for your valuable feedback! We would like to provide additional experimental results to address your concerns and further support our claims.
>
> **[1. Results on more datasets]**
>
> Thank you for the suggestion to incorporate additional datasets to prove generalization. As such, we have extended SparseRefine to the Pascal VOC segmentation dataset on HRNet-W48 in the table below. We obtain **1.8$\times$**   speedup with **0.4** mIoU improvement.
>
> |       | Input Resolution | Latency(ms) |   mIoU
> | ----------- | ----------- | ----- |----- |
> | HRNet-W48 | 512 | 14.7 | 77.8 |
> | HRNet-W48 | 256 | 5.0 | 77.2 |
> | + SparseRefine (Ours) | 512 | 8.1 | 78.2 |
>
> **[2. Describe evaluation metrics in Table 2]**
>
> The metrics in Table 2 are the same as in Table 1. The latency is the end-to-end inference time of the model. The mIoU is the mean Intersection over Union. It measures the overall performance of a segmentation model by calculating the average intersection over union (IoU) across different classes or categories. For intersection, it refers to the pixels that are correctly classified as part of a specific class in both the ground truth and predicted segmentation mask. For union,  it represents the total number of pixels that are classified as part of a specific class in either the ground truth or predicted segmentation mask. Intersection over Union is calculated by dividing the intersection of a class by the union of that class.
>
> **[3. How does the model without ensembler performs]**
>
> As shown in Table 5(c), the direct strategy refers to no ensembler, and the result is 77.7 which shows the effectiveness of ensembler.
>
> **[4. How does the entropy selector behaves on other datasets]**
>
> We have also measured the recall ratio on three other datasets, and the results consistently indicate a similar phenomenon: achieving a high recall ratio with a relatively small density. Below we show the density around 80% recall ratio.
>
> |       | BDD100K | ISIC |   DeepGlobe
> | ----------- | ----------- | ----- |----- |
> | Density | 13.2% | 17.9% | 19.0% |
> | Recall | 83.3% | 80.7% | 80.5% |
>
> **[5. Why final prediction or intermediate feature doesn’t improve performance]**
>
> We hypothesize that the learned features on pixels with high entropy may not be informative and could even contain misleading information. Therefore, incorporating these features would not contribute additional information.
>
> **[6. Is the recall-precision curve based on the sample image or the cityscapes dataset as a whole]**
>
> Based on the Cityscapes dataset as a whole.
>
> **We hope our response has resolved all of your concerns. Please do not hesitate to contact us if there are other clarifications or experiments we can offer. Thank you!**

---

> > ### Comment · Reviewer_Jr1p · 2023-11-22
> >
> > I would like to greatly thank the authors for their work and responses. I have also thoroughly read through all comments of reviewers and authors.
> >
> > Most of the reviewers are concerned about the entropy selector and ensemble, and the authors have provided more experimental results to clarify. My questions of this paper are clarified.
> >
> > Adding refinement modules to models is a popular and efficient way to speed up, and the reported improvements in Latency. However, the significant increase of parameters may be a trade-off to the proposed method. I will think twice of my ratings after further discussion with other reviewers.

---

> > > ### Author Response · Authors · 2023-11-22
> > >
> > > Thank you for taking time to review our rebuttal. We sincerely appreciate your positive feedback and recognition of the extensive experiments conducted. In response to your concern regarding the increase in parameters, we think that the model size is typically not an issue for deploying these vision models. Additionally, it is worth noting that our refinement module only consists of 19.8M parameters, which introduces a relatively small parameter overhead compared to the base model.

---

> ### Author Response · Authors · 2023-11-22
> **Would you give us a response?**
>
> Dear Reviewer Jr1p:
>
> We would like to express our sincere appreciation for your constructive comments and invaluable suggestions again. We have taken all your concerns into careful consideration and addressed them comprehensively in the above rebuttal. Could you help take a look and see whether your concerns are well addressed? If there are any remaining questions, please do not hesitate to let us know. We are more than happy to engage in further discussion and provide any additional information or clarification needed. Thank you very much and look forward to your reply!

---

### Official Review · Reviewer_c8GE · 2023-10-20

**Soundness:** 3 good
**Presentation:** 3 good
**Contribution:** 3 good
**Rating:** 6
**Confidence:** 4

**Summary:**

The manuscript presents an approach to accelerate semantic segmentation inference. The proposed approach combines low-resolution predictions from some standard baseline with sparse high-resolution predictions delivered by MinkowskiUNet. The sparsity is enforced by only looking at pixels with high entropy baseline predictions. These pixels are processed by the sparse feature extractor (Minkowski UNet) while preserving the same sparsity pattern throughout the model. Sparse features are converted to predictions through projection onto the Cityscapes taxonomy. Finally, the joint predictions are recovered by ensembling the dense low-resolution predictions with sparse high-resolution predictions according to regressed weights.

**Strengths:**

S1. The proposed method succeeds to improve the inference speed (1.5x - 2.0x) of popular heavy-weight models while keeping the mIoU performance.

S2. Sparse feature extraction appears as a powerful and under-researched computer vision technique.

S3. Simplicity of the method will likely lead to derivative future work.

S4. I was really surprised that looking at sparse pixels with so little context could contribute that much to the final performance.

S5. I was also surprised that showing low resolution features to the sparse feature extractor did not help.

**Weaknesses:**

W1. The three components of the solution (entropy-based uncertainty, Minkowski engine, weighted ensembes) have been proposed in the related work.

W2. Proper validation of hyper-parameter \alpha has not been discussed (validating on test data is not acceptable),

W3. Training the sparse feature extractor requires a lot of computational power (96 RTXA6000 days).

**Questions:**

Questions

Q1 It would be interesting to ablate capacity of the sparse feature extractor (eg. halving the numbers of feature maps throughout the model).

Q2 Was MinkowskiUNet pre-trained or trained from scratch?

Q3. Show the accuracy for 100% density in Table 5a.

Q4. Include simple average ensembling in Table 5c.

Q5. Explain the magnitude selector.

Q6. Show MACs in Table 7.

Q7 Report minimal hardware requirements (total GPU RAM) for reproducing the experiments

Consider citing earlier work on multi-resolution semantic segmentation:
- HRDA: Context-Aware High-Resolution Domain-Adaptive Semantic Segmentation (ECCV 2022)
- Efficient semantic segmentation with pyramidal fusion (PR 2021)
- HookNet: Multi-resolution convolutional neural networks for semantic segmentation in histopathology whole-slide images (MIA 2021)

---

> ### Author Response · Authors · 2023-11-18
>
> Thank you so much for the positive rating and insightful comments. Your valuable suggestions are very helpful for further strengthening our paper.
>
> **[1. The three components have been proposed in the related work]**
>
> The core contribution of our paper lies in the introduction of a new paradigm – "dense prediction, followed by sparse refinement" – which paves the way for efficient, high-resolution semantic segmentation. We incorporate existing modules to provide one possible instantiation of the new paradigm, but this should not be mistaken as a lack of novelty. Our focus isn't on the design of any specific module, but rather on the novel sparse refinement paradigm itself. Each component can be readily substituted to create a completely new implementation.
>
> **[2. Proper validation of hyper-parameter $\alpha$]**
>
> We ensured that the test data was not utilized during the selection process of α. Our guiding principle for α selection is based on the desired recall ratio. We consistently aim for a recall ratio of approximately 80 percent, which is sufficient to achieve comparable results. Of course, a higher recall ratio can yield improved outcomes, but it also leads to increased running time.
>
> **[3.Training time]**
>
> You may misread the paragraph. The training time is 96 RTX A6000 hours, not days.
>
> **[4. Capacity of the sparse feature extractor]**
>
> We experimented with different model capacities and the results are shown in the following table. Specifically, we incrementally increase the capacity of MinkUNet by expanding channels and adding more stages. The results indicate that performance improves as the model becomes larger, but eventually reaches saturation at 80.9 mIoU.
>
> |       | #of Channels(32$\times$) |   mIoU
> | ----------- | ----------- | ----- |
> | MinkUNet | {1,2,4,8} | 80.5 |
> | MinkUNet | {1,2,4,8,16} | 80.9 |
> | MinkUNet | {1,2,4,8,12,16,24,32} | 80.9 |
>
> **[5. Was MinkowskiUNet pre-trained or trained from scratch]**
>
> We train it from scratch.
>
> **[6. Show the accuracy for 100% density in Table 5a]**
>
> The result is 80.5 when incorporating all the pixels, lower than 80.9 we got with 12% density. This is reasonable because our goal is to refine the pixels that are "difficult to learn in the low-resolution" ones. However, incorporating all the pixels would also involve including many pixels that are easy to learn. This can serve as a shortcut for the model to easily achieve low loss with weaker capability.
>
> **[7. Include simple average ensembling]**
>
> The result is 80.4 when using simple average ensembling, showing the effectiveness of our ensembler.
>
> **[8. Explain the magnitude selector.]**
>
> The magnitude selector calculates the L2 magnitude on the output of the last layer, just before the segmentation head, in order to obtain an importance score for each pixel. This approach is commonly employed in token pruning works to identify and remove unimportant areas. However, the unsatisfactory results obtained from the magnitude selector highlight the lack of a strong correlation between importance and uncertainty.
>
> **[9. Show MACs in Table 7]**
>
> The #MACs for the cuBLAS backend remains the same as the high-resolution model due to the dense input. And the #MACs for SpConv and TorchSparse are the same, shown in the main table. For instance, in the case of HRNet-W48 on Cityscapes dataset, the cuBLAS backend has #MACs of 0.75T, while SpConv and TorchSparse have #MACs of 0.32T.
>
> **[10. Report minimal hardware requirements]**
>
> The GPU RAM required for the task is approximately 20G per GPU, making it compatible with GPUs such as the GeForce RTX 3090 equipped with 24G RAM.
>
> **[11. Consider citing earlier work]**
>
> Thanks! We will cite these works in our revision.
>
> **We hope our response has resolved all of your concerns. Please do not hesitate to contact us if there are other clarifications or experiments we can offer. Thank you!**

---

> > ### Comment · Reviewer_c8GE · 2023-11-23
> >
> > I would like to thank the authors for their exhaustive feedback. My understanding of the presented work has increased. I am looking forward to the discussion with the other reviewers!

---

> ### Author Response · Authors · 2023-11-22
> **Would you give us a response?**
>
> Dear Reviewer c8GE:
>
> We would like to express our sincere appreciation for your constructive comments and invaluable suggestions again. We have taken all your concerns into careful consideration and addressed them comprehensively in the above rebuttal. Could you help take a look and see whether your concerns are well addressed? If there are any remaining questions, please do not hesitate to let us know. We are more than happy to engage in further discussion and provide any additional information or clarification needed. Thank you very much and look forward to your reply!

---

### Official Review · Reviewer_7yKn · 2023-11-01

**Soundness:** 3 good
**Presentation:** 3 good
**Contribution:** 2 fair
**Rating:** 5
**Confidence:** 4

**Summary:**

This paper introduces an efficient method for 2D image segmentation task, without sacrificing the accuracy. A solution is proposed to sparsely refine the interpolated coarse prediction, based on the unconfident predicted regions identified by an entropy selector. The experiments are evaluated on Cityscapes, BDD100K, DeepGlobe and ISIC datasets, validating the efficiency of the proposed method.

**Strengths:**

This work explores applying a sparse refinement on the interpolated coarse prediction, which uses an entropy selector to help to sparsely identify the erroneous regions, without the need to refine the prediction in a full image-size. Thus, this approach gives a reduction in computation during inference.

**Weaknesses:**

1. I agree that the integration of multiple components into a feasible solution is a non-trivial task. However, the composition of such existing works implies that the proposed work lacks sufficient novelties.
2. Although the authors claim the proposed work provides a significant speedup in inference. However, a comparison in terms of a more persuasive metric, GFLOPS, is missing, which is independent of the machine speed and commonly used for measuring the inference efficiency of a network model.
3. The details of the entropy selector, and the elaboration on the effectiveness of the entropy selector are missing.

**Questions:**

Another important approach about the uncertainty in segmentation, e.g. Bayesian Deep Learning method [1], is missing in literature and comparison.
[1] Kendall, Alex, and Yarin Gal. "What uncertainties do we need in Bayesian deep learning for computer vision?." Advances in neural information processing systems (2017).

---

> ### Author Response · Authors · 2023-11-18
>
> Thank you for your valuable feedback! We want to address your concerns about novelty and provide additional clarification and experimental results.
>
> **[1. Lack sufficient novelty]**
>
> Thank you for acknowledging that integrating multiple components is a non-trivial task. The core contribution of our paper lies in the introduction of a new paradigm – "dense prediction, followed by sparse refinement" – which paves the way for efficient, high-resolution semantic segmentation. We incorporate existing modules to provide one possible instantiation of the new paradigm, but this should not be mistaken as a lack of novelty. Our focus isn't on the design of any specific module, but rather on the novel sparse refinement paradigm itself. Each component can be readily substituted to create a completely new implementation.
>
> **[2. GFLOPS metric]**
>
> Thank you for bringing that up. We have included a similar metric, #MACs, in the main table for reference. There is an approximately 1:2 ratio relationship between GMACs and GFLOPs. For example, we reduced the #MACs of HRNet-W48 on Cityscapes dataset from 0.75T to 0.32T per image, achieving a 2.3 times reduction.
>
> **[3. Details and the effectiveness of the entropy selector]**
>
> As discussed in section 3.2.1 of the main paper, the entropy selector is a non-learnable module. It utilizes the logits derived from the low-resolution module to compute the entropy using the entropy formula. Subsequently, it selects the pixels with an entropy value exceeding a predefined threshold. Figure 3(a) also shows the strong correlation between the entropy map and the error map. The high recall ratio shown in Figure 3(b) demonstrates the effectiveness of the entropy selector, as it allows us to identify a significant portion of misclassified pixels. We have also measured the recall ratio on three other datasets, and the results consistently indicate a similar phenomenon: achieving a high recall ratio with a relatively small density. Below we show the density around 80% recall ratio.
>
> |       | BDD100K | ISIC |   DeepGlobe
> | ----------- | ----------- | ----- |----- |
> | Density | 13.2% | 17.9% | 19.0% |
> | Recall | 83.3% | 80.7% | 80.5% |
>
> **[4. Another important approach about the uncertainty]**
>
> Thank you for bringing this to our attention. We have thoroughly read the mentioned paper and implemented their method using HRNet-W48 on Cityscapes. In order to ensure a fair comparison, we selected a resolution of 768 × 1536, which exhibits similar latency to our approach, as indicated in the table. We incorporated Aleatoric & Epistemic uncertainty into our model. However, we observed a result of 80.0, which is lower than the performance achieved by our approach under the same latency. This shows that refinement is the key to our success.
>
> |       | Latency(ms) | mIoU |
> | ----------- | ----------- | ----- |
> | HRNet-W48(768 × 1536) | 30.0 | 79.7 |
> | HRNet-W48(768 × 1536) + Aleatoric & Epistemic uncertainty | 30.0 | 80.0 |
> | HRNet-W48(512 × 1024) + SparseRefine | 30.3 | 80.9 |
>
> **We hope our response has resolved all of your concerns. Please do not hesitate to contact us if there are other clarifications or experiments we can offer. Thank you!**

---

> ### Author Response · Authors · 2023-11-22
> **Would you give us a response?**
>
> Dear Reviewer 7yKn:
>
> We would like to express our sincere appreciation for your constructive comments and invaluable suggestions again. We have taken all your concerns into careful consideration and addressed them comprehensively in the above rebuttal. Could you help take a look and see whether your concerns are well addressed? If there are any remaining questions, please do not hesitate to let us know. We are more than happy to engage in further discussion and provide any additional information or clarification needed. Thank you very much and look forward to your reply!

---

> ### Author Response · Authors · 2023-11-23
> **Deadline coming. Looking forward to your feedback.**
>
> Dear Reviewer 7yKn:
>
> As the rebuttal deadline is less than 8 hours away, we kindly request your feedback. We sincerely hope that our previous feedback adequately addressed your concerns. We would like to express our gratitude once again for your time and previous review, and we eagerly anticipate further discussions!

---

### Official Review · Reviewer_pDrt · 2023-11-01

**Soundness:** 4 excellent
**Presentation:** 4 excellent
**Contribution:** 3 good
**Rating:** 6
**Confidence:** 4

**Summary:**

The paper tackles the problem of efficient semantic segmentation for high-resolution images. The key idea is to perform inference on low-resolution and refine the high entropy pixels using a sparse convolution network in high resolution. Refinement on sparse pixels makes the method efficient compared to the vanilla high resolution semantic segmentation. The proposed method makes significant speedups on Cityscapes dataset on 4 recent neural network architectures maintaining performance.

**Strengths:**

* The idea is simple and makes sense. The area to be refined is indeed sparse, and using sparse NN to the refined area makes sense and should improve the time-complexity.

* I enjoyed the generality of the method. Because the method does not assume any restrictions on the segmentation architecture and only uses the segmentation logit, the method is applicable to any segmentation model. The segmentation model can be plug-and-play.

* The experiments are well conducted. The authors show the generality of the method with 4 recent architectures and on various datasets. I believe there is a computational gain to the method maintaining the performance.

* The paper is well-written and easy to follow. All the expectations made in the introduction are satisfied.

**Weaknesses:**

* I’m not sure how the training data for the refinement was created. To train the refinement module, sparse high entropy pixels are required. How are the high entropy pixels acquired? Is it acquired from the pretrained segmentation architectures? Also, is the refinement model trained for each of the NN architectures in Table 1, or is it universal?

**Questions:**

* (minor) Have the authors explored refining in a multiresolution fashion. It seems the method is applicable to different resolutions, such as 4x downsampling and upsampling twice with 2 refinement architectures.

---

> ### Author Response · Authors · 2023-11-18
>
> Thank you so much for the positive rating and insightful comments. Your valuable suggestions are very helpful for further strengthening our paper.
>
> **[1.How the training data for refinement was created]**
>
> We first get low-resolution logits from the low-resolution model(it has the same architecture as the high-resolution model and we train it with downsampled images). Then we upsample the low-resolution logits to get logits matching the original input resolution(with nearest neighbor interpolation). And sparse pixels with high entropy are calculated from the upsampled logits for training.
>
> **[2. Is the refinement module universal]**
>
> The refinement model is not universal, as the output logits vary across different model architectures, resulting in different selected sparse pixels. Therefore, we trained each neural network architecture based on their respective low-resolution output logits to achieve the optimal results.
>
> **[3. Refine in a multiresolution fashion]**
>
> Thanks for your interesting idea. We carefully considered your idea but found that there may be some difficulties.
>
> Firstly, the performance of the 4x downsampling resolution is notably worse than that of the 2x downsampling resolution as shown in the following table(HRNet-W48 on Cityscapes).  Starting from such a lower performance point poses greater challenges for the refinement module. As far as we know, so far there are no methods able to achieve ~8 points improvements at the level of 70 mIoU.
>
> Secondly, it is worth mentioning that the process of upsampling twice necessitates training two refinement modules. This requirement increases the training effort and diminishes the elegance of the method.
>
> |  | Input Resolution  |   mIoU
> | ----------- | ----------- | ----- |
> | HRNet-W48 | 1024×2048  | 80.7 |
> | HRNet-W48 | 512×1024 | 79.2 |
> | HRNet-W48 | 256×512 | 72.4 |
>
> **We hope our response has resolved all of your concerns. Please do not hesitate to contact us if there are other clarifications or experiments we can offer. Thank you!**

---

> > ### Comment · Reviewer_pDrt · 2023-11-21
> > **Additional questions**
> >
> > I really appreciate the detailed response from the authors. I have further questions and comments regarding the method.
> >
> > 1. How the training data for refinement was created.
> > From my understanding, the low-resolution model is trained from the training data. The goal of obtaining the training data for refinement is to match the distribution of high-entropy areas for validation and split. If the low-resolution model is trained from training data, and produces a high-entropy area on training data, then won't the distribution of the high-entropy model for validation and training have different distributions?
> >
> > 2. It would be interesting to report the scores with universal module. Although the performance would not be optimal, I think the community can benefit from using the kitchen-sink model to quickly see how the model can benefit from using it.

---

> > > ### Author Response · Authors · 2023-11-22
> > >
> > > Thank you for taking time to review our rebuttal. We sincerely appreciate your feedback. Please find below our additional responses to your questions.
> > >
> > > **[1]** Yes, the low-resolution model is trained using the training data. It is important to highlight that we do not manipulate the logits during the training process of the low-resolution model. Once we obtain the low-resolution model, we simply perform inference on all the images using this model to obtain the logits then the high-entropy pixels. Consequently, as long as the distribution of the original images remains similar (which is ensured by the official data split), the distribution of high-entropy areas should also be similar between the training and test data. To provide further confirmation, we conducted an analysis of the distribution of high-entropy areas selected by our method in both the training and test data. The results, presented in the following table, exhibit a similar distribution. The numbers in the table represent the proportion of each class within the selected high-entropy pixels.
> > >
> > > |  | road  |   sidewalk | building | wall | fence | pole | traffic light | traffic sign | vegetation | terrain | sky | person | rider | car | truck | bus | train | motorcycle | bicycle |
> > > | ----------- | ----------- | ----- |----- |----- |----- |----- |----- |----- |----- |----- |----- |----- |----- |----- |----- |----- |----- |----- |----- |
> > > | Train | 8.4% | 9.3%|  23.2% |  3.1% |  3.4% |  8.6% | 1.0% | 2.3% | 20.1% | 3.2% | 2.7% | 4.0% | 1.2% |  5.0% | 0.6% | 0.6% |  0.2% | 0.4% | 2.9%
> > > | Test | 8.5% |  9.2% | 23.2% |  1.4% |  2.5% |  8.9% | 1.3% |  2.2% |  19.9% |  3.1%|  2.7% |  4.8% | 1.1% |  7.2% |  0.4% | 0.4% |  0.3% | 0.5% |  2.3%
> > >
> > > **[2]** Defining and training a universal refinement module is challenging due to the different logits and entropy thresholds used for different base models. In order to assess the transferability across models, we employed the refinement module trained for HRNet-W48 and applied it to refine the outputs of Mask2former-T, SegFormer-B5, and SegNext-L. The results, presented in the table below, align with our expectations, demonstrating a worse performance compared with one refinement module for one base model.
> > >
> > > |  | refinement module trained for HRNet-W48  |  refinement module trained for own use  |
> > > | ----------- | ----------- | ----- |
> > > | Mask2former-T | 71.3 | 81.3 |
> > > | SegFormer-B5 | 76.3 |  81.2 |
> > > | SegNext-L | 78.7 |  82.8 |
> > >
> > > **We hope our response has resolved your questions. If you have further questions, we would be more than happy to engage in further discussion. Thank you very much!**

---

### Meta-Review · Area_Chair_ifrx · 2023-12-10

**Metareview:**

The authors propose a sparse refinement for efficient semantic segmentation of high-resolution images. The bulk of the inference is on low-resolution and the high entropy pixels are then refined with a sparse refinement using a sparse convnet in high-res. The approach is validated on Cityscapes and 4 recent neural architectures. The performance is maintained while large speedups are reported.

- Reviewer pDrt finds unclear the high entropy pixels selection process, the creation of training data and the generalization
- Reviewer 7yKn finds that "the proposed work lacks sufficient novelties", missing GFLOPS comparison results, and missing details of the entropy selector.
- Reviewer c8GE points out that "the three components of the solution (entropy-based uncertainty, Minkowski engine, weighted ensembes) have been proposed in the related work", and "proper validation of hyper-parameter \alpha has not been discussed (validating on test data is not acceptable)".
- Reviewer Jr1p finds the main weakness to be the lack of comprehensive experiments and that "the significant increase of parameters may be a trade-off to the proposed method" and asks clarifications wrt entropy selector and ensemble.

The authors provided responses to all the reviewers, however none of the reviewers is championing the acceptance of the paper and the final ratings are borderline (6,5,6,5).

The meta-reviewer, after carefully checking the reviews, the discussions, and the paper, agree that the paper while reporting significant speedups, also requires a major revision as it lacks in aspects pointed out by the reviewers, like novelty, clarity and details, and sufficient experiments.

The authors are invited to benefit from the received feedback and further improve their work.

**Justification For Why Not Higher Score:**

The paper does not meet the acceptance bar due to the multiple weaknesses and issues pointed out by the reviewers that require a major revision.

**Justification For Why Not Lower Score:**

N/A

---

### Decision · Program_Chairs · 2024-01-16

Reject